# Protective Effect of l-Hexaguluroic Acid Hexasodium Salt on UVA-Induced Photo-Aging in HaCaT Cells

**DOI:** 10.3390/ijms21041201

**Published:** 2020-02-11

**Authors:** Qiong Li, Donghui Bai, Ling Qin, Meng Shao, Xi Liu, Shuai Zhang, Chengxiu Yan, Guangli Yu, Jiejie Hao

**Affiliations:** 1Key Laboratory of Marine Drugs, Ministry of Education, School of Medicine and Pharmacy, Ocean University of China, Qingdao 266003, China; liqiong3360@163.com (Q.L.); 17806250053@163.com (D.B.); QinlingOUC@163.com (L.Q.); betty0301@126.com (M.S.); liuxicsu1992@163.com (X.L.); m13695312569@163.com (S.Z.); Dyan2221@163.com (C.Y.); 2Shandong Provincial Key Laboratory of Glycoscience and Glycotechnology, School of Medicine and Pharmacy, Ocean University of China, Qingdao 266003, China; 3Laboratory for Marine Drugs and Bioproducts, Pilot National Laboratory for Marine Science and Technology (Qingdao), Qingdao 266237, China

**Keywords:** α-l-hexaguluroic acid hexasodium salt, HaCaT cells, UVA-induced photo-aging, mitochondrial dysfunction

## Abstract

This study aimed to show the α-l-Hexaguluroic acid hexasodium salt (G6) protective effect against UVA-induced photoaging of human keratinocyte cells. We found that G6 localized to the mitochondria and improved mitochondrial functions. G6 increased respiratory chain complex activities, which led to increased cellular ATP content and NAD^+^/NADH ratio. Thus, G6 alleviated the oxidative stress state in UVA-irradiated cells. Moreover, G6 can regulate the SIRT1/pGC-1α pathway, which enhanced the cells’ viability and mitochondria energy metabolism. Notably, the anti-photoaging potential of G6 was directly associated with the increased level of MMP and SIRT1, which was followed by the upregulation of *pGC-1α*, *D-LOOP*, and *Mt-TFA*, and with the transcriptional activation of *NRF1/NRF2*. Taking all of the results together, we conclude that G6 could protect HaCaT cells from UVA-induced photo-aging via the regulation of mitochondria energy metabolism and its downstream signaling pathways.

## 1. Introduction

It is well known that repeated exposure to ultraviolet (UV) radiation contributes to photo-aging of the skin. The hypertrophy of the epidermis and occasional hyperkeratosis of the stratum corneum are involved in the histological features of photo-aging [1,2]. Indeed, an increase in the thickness of the basement membrane and an irregular distribution of melanocytes along the basement membrane were also observed [3,4]. Depending on their wavelengths, ultraviolet rays enter the skin to different degrees and interact with skin cells [3,4,5]. Among the ultraviolet rays, UVA (320–400 nm) can penetrate the epidermis and reach the dermis layer, which mainly causes skin photo-aging [6,7,8]. UVA radiation acts on melanocytes and other skin cells, such as keratinocytes, to cause DNA damage through oxidative stress and the production of reactive oxygen species (ROS), which activate signaling pathways associated with cell and tissue growth, differentiation, senescence, and photo-aging [9,10,11]. Moreover, ROS produced in UVA-irradiated human skin cells are primarily responsible for photo-aging [1,12].

The ROS produced in the mitochondria account for almost 90% of the total ROS production in cells [12,13]. Even though the process of oxidative phosphorylation is effective during normal breathing, a small fraction of electrons may “leak” from the electron transport chain (ETC), especially from complexes I and III reduce oxygen to form ROS [13]. Other mitochondrial proteins, such as alpha-glycerophosphate dehydrogenase, alpha-ketoglutarate, and pyruvate dehydrogenase, have been shown to play a role in ROS production [14]. The mitochondria are also the only organelle in animal cells that have their own mtDNA, which is located physically close to the mitochondrial respiratory chain (MRC). Moreover, those point mutations and deletions in mtDNA were reported to accumulate more and more in various tissues during aging [15], which might consequently lead to the declined MRC capacity in various tissues, such as the skeletal muscle and liver [16,17]. In addition, the fact that the mitochondrial ETC is the major ROS production site results in mitochondria being the primary target for oxidative damage. Related studies have shown that with age, mitochondria become larger, decrease in the number, accumulate vacuoles, sputum abnormalities, and experience mitochondrial rupture [18,19].

More importantly, oxidative damage of protein and mtDNA, accompanied by mtDNA mutations, could result in a decrease in mitochondrial respiratory chain enzyme activities, mitochondrial membrane potential, and ATP production [19]. It has been demonstrated that creatine enhances DNA repair caused by UV-induced stress exposure in skin models [20]. Notably, in human clinical settings, formulations containing creatine, acetyl l-carnitine, and NADH can reduce the appearance of aging [21]. The researchers believe that the repair is due to the increased availability of ATP provided by creatine. Under the pressure of ultraviolet light, cells can use this additional ATP to synthesize the required repair enzymes [20,21].

Meanwhile, ultraviolet irradiation can also induce the expression of the ROS production, leading to skin inflammation and downregulation of SIRT1 [22]. It has been reported that an enhancement in sirtuin genes and pGC-1α can increase the activities of complexes I and IV, which reduces oxidative stress [23]. SIRT1 is a nuclear protein belonging to the family of nicotinamide adenine dinucleotide dependent enzymes. Moreover, SIRT1 plays a key role in regulating cell growth, proliferation, and senescence. Therefore, SIRT1 inhibition is generally thought to be a hallmark of aging [24,25], and SIRT1 is involved in UV-induced DNA damage repair, cell metabolism, and photo-aging [26]. A growing body of evidence supports the hypothesis that photo-aging is regulated by a sustained crosstalk between ROS and SIRT1 [27,28]. Moreover, some reports demonstrated that SIRT1 becomes active to protect cells from oxidative stress and reduce ROS production [28,29,30,31].

Marine alginate is a natural anionic polysaccharide extracted from brown algae such as *Laminaria japonica*. It consists of α-L-guluronic acid (G) and β-D-mannuronic acid (M) that occur in homopolymeric M blocks (M-blocks), heteropolymeric random MG blocks (MG-blocks), and homopolymeric G blocks (M-blocks). It has been widely studied and applied in chemical industry, medicinal technology, and food engineering [31]. Moreover, alginate oligosaccharides produced by a variety of degradation methods exhibit excellent biomedical activities including anti-inflammatory activity, antioxidant activity, and neuron protection effects [32,33,34]. However, the protective effect of α-l-Hexaguluroic acid hexasodium salt (G6, Figure 1) is rarely reported. The present study evaluated the protective effect of G6 on photo-aging of HaCaT cells induced by UVA and its mechanism of action. According to our research, G6 is a purely natural ingredient with no toxic side effects. It has high potential to be developed into anti-photo-aging drugs or functional foods with good development and application prospects.

## 2. Results

### 2.1. G6 Inhibits UVA-Irradiated Cytotoxicity in HaCaT Cells

The effects of UVA and G6 on HaCaT cells viability were detected by MTT assays. Additionally, the control group was normalized to be considered as 100%. The 10 μM (G1–G8)-pretreated were significantly protected HaCaT cells from the irradiation of UVA (Figure 2a). In addition, the cells’ viability after G6 treatment was not significantly different from those in the control group, indicating that G6 up to 600 μM had no significant cytotoxicity (Figure 2b). Moreover, G6 (2, 10, 50, 200, and 400 μM)-pretreated were significantly protected HaCaT cells from the irradiation of UVA (Figure 2c).

### 2.2. Effect of G6 on Mitochondrial Membrane Potential (MMP)

We performed a JC-1 assay to determine changes in mitochondrial membrane potential induced by different doses of G6. Figure 3 shows that 30 J/cm^2^ of UVA significantly induced a significant decrease in MMP in HaCaT cells. As shown in Figure 3, pretreatment of HaCaT cells with different concentrations of G6 significantly reversed the decrease in ΔΨm in HaCaT cells induced by UVA as determined using JC-1 assay. At a dose of 2–600 μM of G6 tends to increase the mitochondrial membrane potential. Moreover, the concentration of 2, 10, and 200 μM is preferred. For all subsequent experiments, the G6 (10 μM) was selected.

### 2.3. G6 Co-Localize with Mitochondria

To verify whether FITC was tagged on G6, the reaction product and FITC (as a negative control) were separated on silica gel plate with ddH_2_O:chloroform:methanol (0.1:3:1) and were then observed at 365 nm. As shown in Figure 4a, the fluorescence of the FITC-only group was in the upper part of the plate, and the FITC-G6 line was only in the bottom of the silica gel plate. Our results suggested that G6 was successfully labeled with FITC. Additionally, we detected the compounds with the aniline:diphenylamine:phosphoric acid reagent. Then, this compound was used to detect the localization of G6 in HaCaT cells.

The co-localization studies were performed to identify the intracellular localization of G6 in HaCaT cells using confocal fluorescence microscopy. Some researchers have shown that the dysfunction of mitochondria participates in processes of UVA-induced photo-aging [32,33,34,35,36], which motivated us to investigate the presence of G6 in the mitochondria using MitoTracker Red CMXRos. As shown in Figure 4b, incubation with MitoTracker Red CMXRos (red) and FITC-labeled G6 (green) resulted in some punctuated staining patterns of orange-yellow fluorescence when the respective channels were merged. Our results suggest that G6 and mitochondria were co-localized. Therefore, G6 was primarily localized in mitochondria. Moreover, we found that G6 has a direct interaction with mitochondria, which directly affect mitochondrial functions.

### 2.4. Effect of G6 on ROS Production, ATP Content, and NAD^+^/NADH Ratio Change

As shown in Figure 2 and Figure 3, the optimal concentrations of G6 for protecting HaCaT cells from UVA irradiation were 2, 10, and 200 μM. Figure 5a shows that 10 μM of G6 markedly reduced ROS levels. Thus, to reach sufficiently maximal protection of G6, we used 10 μM treatment for 48 h in the following tests. Meanwhile, Figure 5b indicated that UVA (30 J/cm^2^) significantly reduced ATP levels. As expected, pretreatment with 10 μM G6 significantly increased ATP levels in UVA-induced HaCaT cells. These results indicated that the pretreatment of 10 μM G6 was the most effective in protecting UVA-irradiated HaCaT cells. Additionally, we continued to measure the changes of NAD^+^/NADH ratio in HaCaT cells. Figure 5c shows that the intracellular NAD^+^/NADH ratio was markedly decreased, after 30 (J/cm^2^) UVA irradiation of HaCaT cells. Whereas, the addition of 10μM G6 significantly increased the intracellular NAD^+^/NADH ratio. Furthermore, there was a significant difference in the NAD^+^/NADH ratio between the HaCaT cell after 10 μM G6 was added to the normal cells and the control groups. Results of ROS production assay are shown in Figure 5a. Treatment with UVA (30 J/cm^2^) resulted in a significant increase in ROS production in HaCaT cells. Moreover, G6 markedly decreased cellular ROS production of HaCaT cells.

### 2.5. Effect of G6 on Mitochondrial Functions

To explore whether G6 can affect cellular mitochondrial function, we measured the activity of mitochondrial complex I and complex II. Figure 6 shows that 10 μM G6 pretreatment significantly increased the activity of mitochondrial complex I and complex II in HaCaT cells. These results shown in Figure 6 indicated that G6 can significantly enhance mitochondrial function. 

### 2.6. Molecular Docking Predicted the Possible Interaction between G6 and SIRT1 Protein

Molecular docking can predict the possible binding sites of G6 within the active cavity of SIRT1 at SIRT1:G6 binding ratios of 1:1 (Figure 7). As shown in Figure 7, residues of Asp-204, Glu-416/208, Lys-203, and Arg-446 formed hydrogen bonds with one molecule of G6. Furthermore, these residues were also observed in the crystal structure of SIRT1/G6 complex with the 1:1 ratio, which interacted with G6 by hydrogen bonds. The observed results showed that it is likely that G6 binds within the SIRT1 pocket to produce a stable complex. Previous research supports our findings (Figure 7).

### 2.7. Effect of G6 on Key Protein Expressions of SIRT1 Signaling Pathway

We performed Western blot analysis to measure the expression of SIRT1 and pGC-1a. As shown in Figure 8, the level of SIRT1 and pGC-1a proteins were significantly downregulated in the UVA group. After treatment with 10 μM G6, the content of SIRT1 and pGC-1a protein increased. Additionally, the level of SIRT1 and pGC-1a in HaCaT cells were not significantly different from those in the control group (Figure 8a,b). As shown in Figure 8b,c, the level of pGC-1a protein and MMP sharply decreased, meanwhile the expression of ROS increased using 80 mM NAM in HaCaT cells. Consistent with these data, the levels of SIRT1, pGC-1a protein, and MMP markedly decreased, and the content of ROS significantly increased via co-treatment with 10 μM G6 and 80mM NAM in HaCaT cells. Clearly, 10 μM G6 reversed the downregulation of SIRT1 and pGC-1a expression levels in UVA-irradiated HaCaT cells, and SIRT1 protein is the key target.

### 2.8. Effects of G6 on the mRNA Expression of SIRT1 Pathways

It has been confirmed that SIRT1 and pGC-1a promote mitochondrial regeneration and energy metabolism by modulating multiple steps in AMPK/FoXO1 pathways [37]. Additionally, the role of G6 in the SIRT1 signaling pathway was explored using RT-PCR assays. As shown in Figure 9, *SIRT1* mRNA content was markedly increased compared to the control group after 24 h of G6 treatment. Meanwhile, 10 μM G6 co-acted with normal cells, and the expression of *SIRT1* mRNA was not significantly different from the control group. Additionally, in Figure 9, the mRNA expressions of the target gene downstream of the SIRT1 pathway were consistent with SIRT1.

## 3. Discussion

The skin is the main organ to protect the body from noxious substances, such as toxic chemicals and ultraviolet radiation, which lead to skin photo-aging or skin cancer [38]. The main source of UV exposure in humans is ultraviolet A (UVA) rays, which causes the generation of reactive oxygen species (ROS), such as hydrogen peroxide, superoxide anion, and hydrogen peroxide [39]. In addition, ROS are the critical mediators of many of UV-induced biological effects [40,41,42]. The ROS produced after UVA irradiation induce direct oxidative damage to skin constituents and may also indirectly cause damage to cellular structural proteins, lipids, and polysaccharides [43]. As UVA-induced cell damage is mainly attributed to the harmful effects of free radicals, molecules that can scavenge radicals are primarily promising as radio-protectors.

In our study, the protective ability of G6 and the underlying biology mechanisms against UVA-induced photo-aging in HaCaT cells were revealed. G6 markedly enhanced the activity of pGC-1α by upregulating SIRT1. In addition, as both the activity of *pGC-1α* and the expression levels of *NRF-1*, *NRF-2*, and *ERRα* increased, and the intracellular ROS content decreased. Meanwhile, the activation of *Mt-TFA* and *D-Loop* promoted mitochondrial production and metabolism, and then increased MMP and ATP expression. In this study, we demonstrated that the protective effect of G6 against UVA-induced photo-aging in HaCaT cells is mainly mediated by SIRT1/pGC-1α activation. The results suggest that G6 is an effective component in food supplements for anti-photo-aging and skin protection. Our results suggest that G6 can be used for the preparation of skincare products, such as dermatological creams or lotions, to develop the regeneration and repair of UVA-irradiated skin. UVA exposure leads to oxidative stress by the overexpression of ROS. As a result of the oxidative stress, in spite of the enzymatic and non-enzymatic antioxidant defense systems of the skin that has been damaged, there is oxidative damage of mitochondrial DNA (mtDNA), genomic DNA, proteins, and lipids. Notably, the key target for ROS is mtDNA, as its decline and damage in function lead to vicious cycle-like effects, by further enhancing the ROS content [44].

In this study, we demonstrated that the production of ROS increased, and the viability of HaCaT cells decreased after UVA irradiation, whereas G6 enhanced cell viability and the decrease of ROS generation. Therefore, we revealed that G6 can protect HaCaT cells from UVA-induced oxidative damage. This result was consistent with those of Abbas Mirshafiey et al. (2016), who determined that the small molecule G2013 (C_6_H_10_O_7_) decreases the content of ROS in rats [45]. G2013 was recommended as an anti-aging and preventive agent for a number of diseases related to free radicals [46]. In addition, the ROS produced in mitochondria account for almost 90% of total cellular ROS. The fact that the mitochondrial electron transport chain is the major ROS production site makes mitochondria the key target of oxidative damage. Therefore, the mitochondrial aging theory is associated with the free radical theory [15,47].

The overexpression of ROS results in mtDNA damage and other structures of the cell are also damaged, such as lipids, membranes, and proteins. The mtDNA damage in sites coding for ETC proteins decreases the ATP content, and further increases the ROS content [48,49,50,51]. Notably, the inner mitochondrial membrane (mainly at complexes I and II) is the main target of ROS damage [50,52]. As shown in Figure 3, Figure 4, Figure 5 and Figure 6, our results revealed that UVA irradiation results in a decrease of MMP, ATP content, and complex I/II activities in HaCaT cells. Importantly, the pretreatment with G6 removed the UVA-induced mitochondrial metabolic disorder. Moreover, our present study (Figure 4) showed that G6 and mitochondria colocalize. These findings further suggest that G6 directly interacts with the mitochondria. Furthermore, our observations also showed that G6 is primarily localized in the mitochondria and directly affects the mitochondrial functions (Figure 6). Mitochondrial retrograde signaling starts with several main signals (such as ROS, ADP/ATP, and NAD^+^/NADH ratios), and ROS can lead to aging [53]. As shown in Figure 5b,c, exposure to UVA radiation can reduce ATP generation and NAD^+^/NADH ratio. The pretreatment with G6 significantly reversed the UVA-induced decrease of ATP content and NAD^+^/NADH ratio, to the extent that the levels of these molecules were close to those of non-irradiated cells.

In addition, these retrograde messengers activate cytosolic transducers by an oxidative modification that interact with small molecules, including Ca^2+^, NAD^+^, and AMP, which modulate the activities of certain transcription factors. These include pGC-1a, SIRT1, mTOR, and CREB, which affect the mitochondrial and cellular function, namely the energy supply, and then change the progress of cellular senescence and proliferation [54,55,56]. SIRT1 is a member of the family of sirtuins that catalyze the deacetylation of various substrates by utilizing nicotinamide (NAD^+^) as a substrate [57]. Furthermore, the NAD^+^/NADH ratio is much higher than control in the case of UVA+G6. Furthermore, it regulates the activity of several substrates, including peroxisome proliferator-activated pGC-1α and p53 in some signaling pathways, such as the Rap1/Ras, PI3K-Akt, MAPK, and longevity regulating pathway [22,58,59]. On the one hand, it has been reported that SIRT1 plays an important role in regulating cellular homeostasis via influencing neuron survival, insulin sensitivity, glucose metabolism, and mitochondrial biogenesis [60,61]. On the other hand, it can increase pGC-1α activity and inhibit ROS production, which improves mitochondrial dysfunction [62]. Those previous works support our findings (Figure 8). In addition, our report demonstrated that G6 regulates mitochondrial biogenesis and metabolism by activating the NADH/SIRT1/pGC-1α signaling pathway in HaCaT cells (Figure 8 and Figure 9). It has been reported that adding nicotinamide riboside or nicotinamide mononucleotide in XPA cells can promote the NAD^+^-SIRT1-pGC-1α via decreasing the PARP effect and then reversing the mitochondrial dysfunction [36]. NRF2 is a key transcription factor regulating the redox balance in skin aging because it is important to activate the antioxidant system and prevent further decrease of ROS in all types of skin cells [63]. Several in vitro studies have confirmed that the upregulation of the NRF2-related pathway can protect keratinocytes and melanocytes from UVB-induced photo-aging [1,64]. Keratinocytes with the *NRF1* gene silenced are sensitive to killing after UVB-irradiation [65,66]. In addition, pGC1α activation is controlled by deacetylation of lysine residues via the deacetylase SIRT1 [67], which promotes mitochondrial biogenesis [68] and controls the generation of mitochondrial transcription factor A (Mt-TFA) [69]. Here, we demonstrated that G6 protects HaCaT cells from UVA-induced photo-aging mainly through activating SIRT1 and its downstream target genes (e.g., *D-LOOP*, *NRF1/NRF2*, *pGC-1α*, *ERRa*, and *Mt-TFA*), as shown in Figure 9.

In summary, our study revealed the molecular mechanism by which G6 protects HaCaT cells from UVA-induced photo-aging: the activation of the antioxidant system by the upregulation of SIRT1/pGC-1α is critical for the protection of skin cells from UVA-induced photo-aging. In vitro experiment results showed G6 had activated the antioxidant system, which indicated the high potential of G6 as a candidate supplement for skin protection or the preparation of skincare products.

## 4. Materials and Methods

### 4.1. Materials and Reagents

G6 (Figure 1), with chemical and physical properties in Table 1, was purchased from Lantai Pharmaceutical Company (Qingdao, China). Fetal bovine serum (FBS), streptomycin, penicillin G, and MEM medium were purchased from Gibco (Grand Island, NY, USA). Anti-SIRT1 (#2496T), anti-β-actin (#4970s), and anti-pGC-1a (#GR3210687-3). All other reagents were purchased from Sigma–Aldrich Chemical Co. (St. Louis, MO, USA).

### 4.2. Cell Culture and Treatments

The human HaCaT cell line was obtained from the American Type Culture Collection (ATCC, Manassas, VA, USA) and cultured in MEM medium supplemented with 10% heat inactivated FBS, 2 mM glutamine, and 1% penicillin/streptomycin. HaCaT cells were used within seven generations. 

Firstly, Human keratinocyte HaCaT cells (2 × 10^4^ cells/mL) were grown in 96-well plates for 12 h. Then, those cells were incubated in serum MEM with or without various concentrations of G6 in a humidified atmosphere of 5% CO_2_ at 37 ℃ for 48 h. The medium was discarded, serum-containing MEM was added and exposed in different doses of UVA irradiation. After the light source was slaked, it was incubated in a cell incubator.

### 4.3. MTT Assay

After the cells were treated for 24 h, the effect of G6 on cell viability was monitored by the MTT assay. MTT solution (5 mg/mL in PBS buffer) was added and incubated for 4 h. Then, the medium was removed and replaced with DMSO to dissolve the potassium salt and the absorbance was measured at 570 nm [70,71].

### 4.4. Measurement of Mitochondrial Membrane Potential (MMP, ΔΨm)

Determination of MMP was carried out using the proportion dye JC-1(5,5′,6,6′-tetrachloro-1,1′,3,3′-tetraethyl-benzimidazolyl-carbocyanine iodide) [72]. After the cells were treated for 24 h, the HaCaT cells were incubated with 10 µg/mL of JC-1 for 45 min at 37 °C, and then analyzed by a dual-wavelength/double-beam recording spectrophotometer (Ex490/Em530 and Ex525/Em590). MMP was calculated as follows: ΔΨm = A525–590/A490–530.

### 4.5. Measurement of Reactive Oxygen Species (ROS) Generation 

The intracellular accumulation of ROS following UVA radiation was detected by fluorescence microscopy using DCFH2-DA. The HaCaT cells (1.5 × 10^5^ cells/well) were cultured in a 12-well plate in MEM supplemented with 10% FBS. Then, the culture medium was renewed when the cells reached 85% confluence. After G6 treatment (2–600 µM) for 24 h and UVA irradiation (30 J/cm^2^), cells were incubated with 10 µM DCFH-DA in the culture medium at 37 °C for 35 min. Then, washed twice with PBS buffer. Finally, the fluorescence was measured at a Nikon TE2000 microscope at Ex535/Em610 (Flex Station 384, Molecular Devices, NY, USA) [73,74].

### 4.6. Determination of ATP Content

HaCaT Cells were cultured in six-well plates. After various treatments, cells were cracked by 0.5% Triton X-100 in 100 mM glycine buffer, pH 7.4. Intracellular ATP levels were assayed with an ATP bio-luminescence assay kit (Sigma, NY, USA) based on the luciferase-catalyzed oxidation of d-luciferin [75].

### 4.7. NAD^+^/NADH Ratio Change

HaCaT cells were seeded in six-well plates at a density of 2 × 10^5^ cells per well. After various treatments, NAD^+^/NADH ratio changes were then determined according to the manufacturer’s standard protocol provided with the employed NAD^+^/NADH assay kit (Abcam, Cambridge, UK).

### 4.8. Activity of Mitochondrial Complexes I and II

HaCaT cells were cultured in 100 mm plates and washed in PBS. After various treatments, cells were fragmented by ultrasonication, and the crude homogenates were used to test complexes I and II activities. NADH–CoQ oxidoreductase (complex I) activity was assayed according to the Kumar’s method [76]. Additionally, succinate–CoQ oxidoreductase (complex II) was assayed by Humphries’s method [77].

### 4.9. Western Blot Analysis

Briefly, HaCaT cells were seeded in six-well plates at a density of 1 × 10^6^ cells per well. After various treatments HaCaT cells were rinsed with PBS and lysed in ice-cold RIPA buffer containing phosphatase inhibitors and protease inhibitor for 45min. Thirty micrograms of protein were separated on 12% SDS-PAGE and transferred to a PVDF membrane (Millipore, Bedford, MA, USA) for Western blotting. Western blot analyses with antibodies against SIRT1, pGC1α, and β-actin were performed as described previously. The blots were captured using enhanced chemiluminescence (Thermo Fisher Scientific, Waltham, MA, USA). Additionally, densitometry analysis was conducted by an Image J software (National Institutes of Health, Bethesda, MD, USA).

### 4.10. Real-Time RT-PCR Assay 

HaCaT cells (1 × 10^6^ cells/well) were treated with G6 (10 μM). After incubation for 24 h the solution was discarded, and then irradiated with UVA (30 J/cm^2^). After that, HaCaT cells were harvested for the preparation of total RNA by FavorPrep Tissue Total RNA Extraction Mini Kit (Favorgen biotech Corp., Taipei, Taiwan), and 10μL of RNA was reverse transcribed into cDNA using ReverTra Ace qPCR RT Master Mix with gDNA (Toyobo, Japan). Finally, target cDNA levels were quantified by RT-PCR using an ABRE PRISM 7500 Sequence Detection System (Applied Biosystems, San Diego, CA, USA) using SYBR green (Toyobo, Japan). The qPCR assay using the following primers: *SIRT1* mRNA, 5′-TCAGTGTCATGGTTCCTTTGC-3′ and 5′-AATCTGCTCCTTTGCCACTCT-3′; *NRF-1* mRNA, 5′-AGCACGGAGTGACCCAAAC-3′ and 5′-AGGATGTCCGAGTCATCATAAGA-3′; *D-LOOP* mRNA, 5′-TATGGAGTGACATAGAGTGTGCT-3′ and 5′-GTCGCTACACCACTTCAATCC-3′; *NRF-2* mRNA, 5′-CAGCGACCTTCGCAAACAAC-3′ and 5′-CATGATGAGCTGTGGACCGT-3′; *pGC-1a* mRNA 5′-TTGCTAAACGACTCCGAGAACA-3′ and 5′-CAACTGACCCAAACATCATACCC-3′; *Mt-TFA* mRNA, 5′-AGCCGCATCGCCGTCTCCTA-3′ and 5′-CAGCGCTGAGTCGGTCACCC-3′; *ERRα* mRNA, 5′-GGGGAGCATCGAGTACAGC-3′ and 5′-AGACGCACACCCTCCTTGA-3′; *β-actin* mRNA, 5′-TAAGGAGAAGCTGTGCTACG-3′ and 5′-ATACTCCTGCTTGCTGATCC-3′. The quantitative PCR reactions was performed with the following program of denaturation at 95 °C for 1 min, annealing at 95 °C for 15 s, and elongation at 60 °C for 45 s. Results were presented as levels of expression relative to those of controls after normalization to *β-actin* using the 2^−ΔΔCt^ method [78].

### 4.11. Preparation of FITC-Labeled G6

The FITC-labeled G6 was prepared according to the published literature with slight modifications [79]. Briefly, G6 (100 mg) were dissolved in 2 mM of 10% hexamethylenediamine acetate solution. Then, 4 mM of NaBH3CN was added to make the molar ratio of NaBH_3_CN:10% hexamethylenediamine acetate:G6 to be approximately 200:100:1, and then stirred for 12 h in the dark. The pH value of the resulting solution was adjusted to 7 with HCl. After that, the reactant was subjected to dialysis (MW 500 Da) and then freeze-dried. The product was re-dissolved in the Na_2_CO_3_/NaHCO_3_ buffer (pH = 9), and then FITC (10 mg) was added. Finally, the mixture was heated for 24 h at 45 °C and subjected to dialysis for another 24 h, and then freeze-dried.

### 4.12. Intracellular Localization of G6

To confirm the intracellular localization of G6, HaCaT cells (1 × 10^4^/mL) were cultured on glass bottom cell culture dishes and incubated with FITC-G6 (10 μM) for 12 h. After being washed three times by PBS, the above cells were incubated with MitoTracker Red CMXRos (50 nM) (Invitrogen, Carlsbad, CA, USA) for 40 min at 37 °C, and cells were observed under a Nikon A1 confocal microscope (Nikon Corporation, Tokyo, Japan). Confocal fluorescent imaging was conducted as follows: FITC-labeled G6, Ex488/Em520; MitoTracker Red CMXRos, Ex578/Em600.

### 4.13. Molecule Docking Studies

Molecular docking was performed usingAMBER10:EHT forcefield and MOE. Briefly, the X-ray crystal structures of the SIRT1 protein (PDB code:5BTR) was downloaded from the protein data bank (http://www.rcsb.org), and was preprocessed by the UCSF ChimeraX software, including the removal of all hetero-atoms and the water molecules. Additionally, compounds G6 were drawn in Chem3D Pro saved as mol2 format, and then minimized using 10,000 steps of steepest minimization in MOE. After that, the docking research used induced fit docking approach, according to the flexibility of the residues’ side chains at the binding site. The produced conformation, which had the best score was selected for the analysis.

### 4.14. SIRT1 Inhibitor Interferes with SIRT1 Protein Expression

The HaCaT cells were pre-incubated with different concentrations of nicotinamide at 37 °C for 12 h. Next, cells were washed with ice-cold PBS and collected by centrifugation. Protein concentrations were measured by a Pierce™ BCA Protein Assay Kit (Thermo Fisher Scientific, Waltham, MA, USA). Additionally, the content of SIRT1 protein was measured as mentioned in Section 4.9 above. Finally, the supernatant was collected and stored at −80 °C until use.

### 4.15. Statistics

All values are expressed as means ± S.E.M. Statistical significance was determined by using one-way ANOVA with Bonferroni’s post hoc tests between the two groups. The criterion for significance was set at *p* < 0.05.

## Figures and Tables

**Figure 1 ijms-21-01201-f001:**
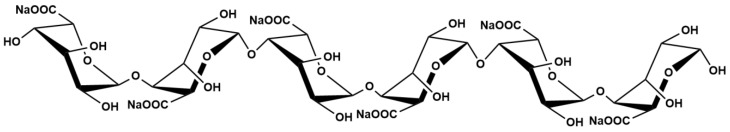
Chemical structure of α-l-Hexaguluroic acid hexasodium salt (G6).

**Figure 2 ijms-21-01201-f002:**
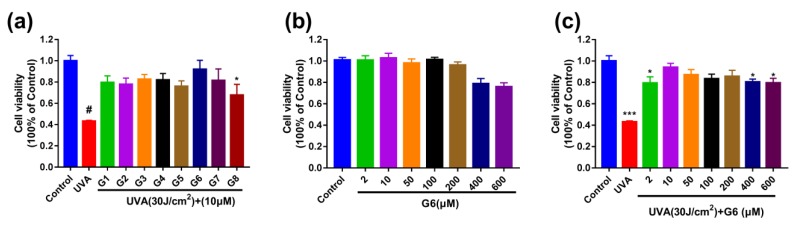
Effects of G6 on cytotoxicity of HaCaT cells determined using MTT assays. (**a**) The HaCaT cells were pretreated with 10 μM G1–G8 for 48 h, and then irradiated with UVA (30 J/cm^2^). (**b**) HaCaT cells were treated with 0, 2, 10, 50, 100, 200, 400, or 600 μM G6 for 48 h. (**c**) The cells were pretreated with 0–600 μM G6 for 48 h, and then irradiated with UVA (30 J/cm^2^). Values are the mean of nine replicates ± S.E.M (*n* = 9). Significant difference compared to the control group, * *p* < 0.05, *** *p* < 0.001 and # *p* < 0.0001

**Figure 3 ijms-21-01201-f003:**
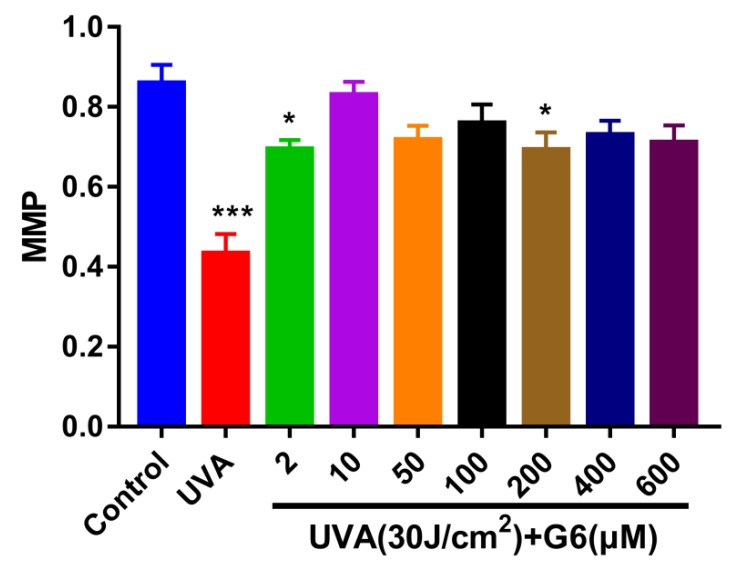
HaCaT Cells were incubated with specific concentrations of G6 for 48 h and then irradiated with UVA (30 J/cm^2^). Significant difference from control group: * *p* < 0.05 and *** *p* < 0.001, bars and error bars are expressed as mean ± S.E.M (*n* = 9).

**Figure 4 ijms-21-01201-f004:**
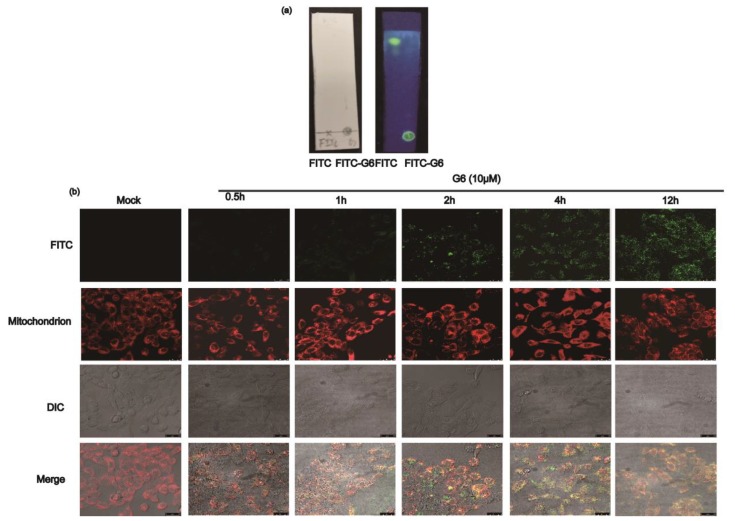
G6 localizes to the mitochondria. (**a**) Thin-layer chromatography results of FITC labeled G6; (**b**) laser-scanning confocal microscopy images of HaCaT cells incubated with MitoTracker Red CMXRos (100×) and FITC-G6. Red fluorescence from MitoTracker Red CMXRos and green fluorescence from FITC-labeled G6, and orange-yellow fluorescence from these two merged sources of fluorescence.

**Figure 5 ijms-21-01201-f005:**
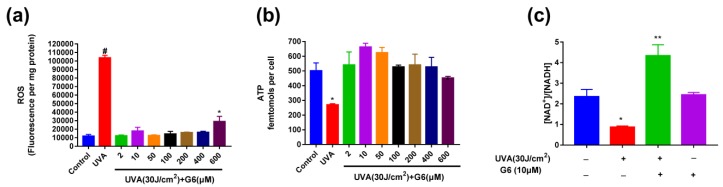
Effect of G6 on ROS production, ATP content, and NAD^+^/NADH ratio change. (**a**) The reactive oxygen species (ROS) production assay using 2 μM MCFH2-DA probe; (**b**) the ATP content; (**c**) the NAD^+^/NADH ratio change. The values are presented as the means ± S.E.M from at least nine independent experiments. Significant difference from control group: * *p* < 0.05, ** *p* < 0.01 and # *p* < 0.0001.

**Figure 6 ijms-21-01201-f006:**
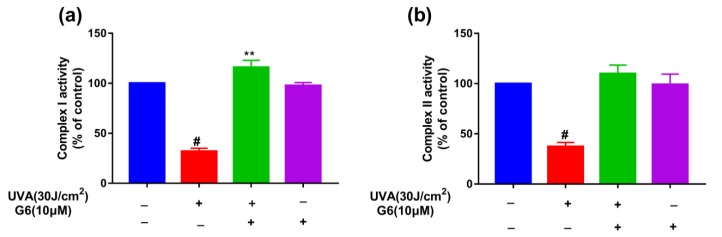
Effect of G6 on mitochondrial functions. (**a**) Activities of complex I; (**b**) activities of complexe II. (blue color, the control group; red color, treated with 30 J/cm^2^ UVA; green color, treated with 30J/cm^2^ UVA and 10 μM G6; black color, treated with 10 μM G6.) The values are presented as the means ± S.E.M from at least three independent experiments. Significant difference from control group: ** *p* < 0.01 and # *p* < 0.0001.

**Figure 7 ijms-21-01201-f007:**
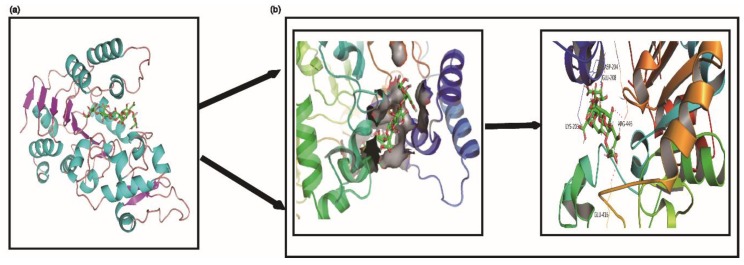
Molecular docking predicted the possible interaction between G6 and SIRT1 protein. (**a**) The electrostatic surface representation; (**b**) cartoon representation of the docking model of G6 binding to human SIRT1, using PyMOL1.8.

**Figure 8 ijms-21-01201-f008:**
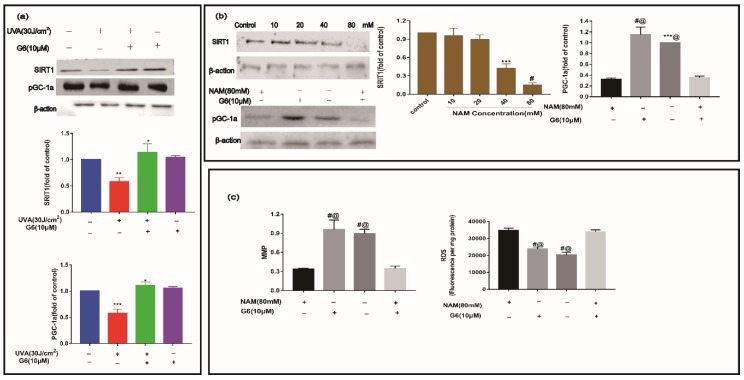
Effect of G6 on SIRT1 and pGC1a protein expression in HaCaT cells. (**a**) HaCaT cells were treated with or without G6 for 48 h, and were then incubated in the presence or absence of UVA. (**b**) Furthermore, HaCaT cells were incubated in the presence or absence of 80 mM NAM, and treated with or without G6 for 48 h. (**c**) The levels of MMP and ROS. Values are the mean ± S.E.M of results from at least three independent experiments, significant difference compared to the control group, *** *p* < 0.001 and # *p* < 0.0001; ***,@ *p* < 0.001 and #,@ *p* < 0.0001, compared with the NAM-treated group, * *p* < 0.05, ** *p* < 0.01.

**Figure 9 ijms-21-01201-f009:**
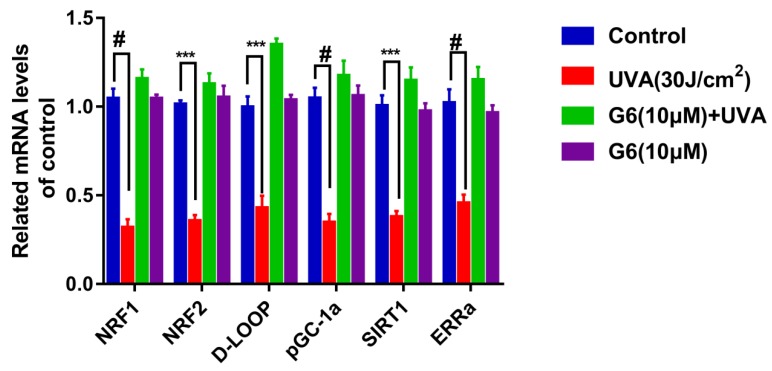
Effect of G6 treatment on mRNA expression of the SIRT1/ pGC-1a pathway in HaCaT cells. HaCaT cells were pretreated with G6 (10 μM) in MEM medium for 24 h. Then, the cells were exposed to UVA (30 J/cm^2^) irradiation. mRNA levels were analyzed by RT-PCR as described in Section 4.13. Values are means ± SEM from six independent experiments, significant difference compared to the control group, # *p* < 0.0001 and *** *p* < 0.001.

**Table 1 ijms-21-01201-t001:** The chemical and physical properties of G6.

Formula	*M*w	Source	Purity	Storage	Properties
C_36_H_44_O_37_Na_6_	1206.65 Da	Marine brown algae	≥96%	0~−20 °C	Off-white to light yellow powdery or flocculent lyophilisate; odorless, non-irritating odor; hygroscopic

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
