# Peer review of "Protective Effect of l-Hexaguluroic Acid Hexasodium Salt on UVA-Induced Photo-Aging in HaCaT Cells"

_ijms, 2020, doi:10.3390/ijms21041201_

Round 1

Reviewer 1 Report

The current work presented by Qiong Li et al is described by using a so called guluronic acid hexaose (C6) to protect the HaCaT cell against UVA-induced photoaging. The author claimed the C6 compound can localized on mitochondria and increased its function. MTT assay, mitochondrial membrane potential, ROS, ATP content, NAD+/NADH ratio assays were reported in the study. The C6 seem to be able to regulate the SIT1/pGC-1ɑ pathway. The mRNA expression of SIRT1/pGC-1a was also discussed.

1. However, the article writing does not meet a scientific logic writing. The most important compound is C6, but author did not give a proper define. It is not suitable to just using C6 in the article title. In the introduction part, it is needed to explain more about G6 compound and the motivation to study G62.

2. In the page 2, author mentions that alginate, consist of ɑ-D-mannuronic and β-L-guluronic aicd, has excellent biomedical activities, which refer to ref 31. However, the ref 31 was talking about “spontaneous coassembly of biologically active nanoparticles via affinity binding of heparin-binding proteins to alginate-sulfate.” Besides, ref 31’s compound is alginate-sulfate. There is almost no connection between ref 31 of current article.

3. It is very confused of C6 compound? It is a monosugar or 6-mer sugar as Figure 1 shown? What is the correct name of this compound? In the abstract uses “Guluronic acid hexaose (G6)”, but in the summary part uses “gurouronic acid, such as G6.” If the G6 means a 6-mers sugar, it is important to give a clear structure define, such as ɑ or β linkage and the link position. In the “4.1 material and regents” part, the G6 information is too ambiguous.

4. In the summary part, “Therefore, gurouronic acid, such as G6, may be a desirable food supplement for skin protection or the preparation of skincare products.” This kind of judge is not suitable. How is the relationship of gurouronic acid can be a food supplement for skin protection? Currently, author used C6 not gurouronic acid. Besides, there is nor any evident experiments to show human digest system of C6 uptake which can relate to skin.

5. Too many grammatical errors. Please refer the article to a native speaker for proof reading. Especially between each specific number and punctuation unit, a space is needed in the all article as well as the figures.

6. Figure 4 (a), the TLC analysis is not a suit way. “ To verify whether FITC was tagged on G6, the reaction product and FITC (as a negative control) were separated on silica gel plate with ddH2O:chloroform: methanol (0.1:3:1) and were then observed at 365 nm. As shown in (Figure 4(a)), the fluorescence of the FITConly group was in the upper part of the plate, and the FITC-G6 line was only in the bottom of the silica gel plate. Our results suggested that G6 were successfully labeled with FITC. Additionally, we detected the compounds with the aniline: diphenylamine: phosphoric acid reagent. Our results shown that FITC-G6 with the complete structure of G6.” At lease, it is needed to have a developed solvent can run TLC analysis and FITC-G6 is not in the bottom part. At least the Rf at 0.1 or 0.2 to show the purity of the FITC-G6. Please show the chemical reaction in the 4.11 part.

7. The resolution of Figure 4 (b) is too poor, the images are not convinced.
Especially, in all of the DIC images, the cell seems not healthy?

8. Figure 5 (c). Why in the case of UVA+G6, the NAD+/NADH ratio is much
higher than control?

9. Discussion part is also very confused. Some descriptions are not refered
to correct figure.

10. More reference is needed. For example:
1) Page 1 line 42. The ROS produced in the mitochondria account for
almost 90% of the total ROS production in cells.

In summary, the paper is not suitable for publication yet in IJMS in its current form. The introduction and discussion part needed to be rewritten. But it can be considered again after major revision.

Author Response

Point 1: (1)It is not suitable to just using C6 in the article title. In the introduction part, it is needed to explain more about G6 compound and the motivation to study G6.(2) There is almost no connection between ref 31 of current article.(3) More reference is needed. For example: Page 1 line 42. The ROS produced in the mitochondria account for almost 90% of the total ROS production in cells.

Response 1: We appreciate the reviewer’s serious consideration of this issue. We have made additions marked in red in the revised manuscript according to your suggestion.

(1) Based on our past experience in the laboratory experiments and platform regarding the research of drugs pharmacology, we screened several  L-guluronic acid (G) activities , and then selected L-Hexaguluroic acid hexasodium salt with better activity for further mechanism research.

(2)“Marine alginate, a kind of acidic linear polysaccharide that consists of b-D-mannuronic acid and a-L-guluronic acid, has been reported widely due to its excellent biomedical activities[31].” was corrected as “Marine alginate is a natural anionic polysaccharide extracted from brown algae such as Laminaria japonica. It consists of  L-guluronic acid (G) and D-mannuronic acid (M) that occur in homopolymeric M blocks(M-blocks), heteropolymeric random MG blocks(MG-blocks),and homopolymeric G blocks(M-blocks). It has been widely  studied and applied in  chemical industry, medicinal technology and food engineering [31]. Moreover, alginate oligosaccharides produced by a variety of degradation methods exhibit excellent biomedical activities including anti-inflammatory activity,anti-oxidant activity and neuron protection effects[32-34]” in line 76.

(3) “The ROS produced in the mitochondria account for almost 90% of the total ROS production in cells” was corrected as “The ROS produced in the mitochondria account for almost 90% of the total ROS production in cells[12.13].”

It’s hoped that our answers could give a clear explanation to your question.

Point 2: In the “4.1 material and regents” part, the G6 information is too ambiguous.

Response 2: Dear reviewer, we are greatly appreciated your professional comments. We have supplemented the chemical and physical properties of G6 in the “4.1 material and regents” part.

Point 3:In the summary part, “Therefore, gurouronic acid, such as G6, may be a desirable food supplement for skin protection or the preparation of skincare products.” This kind of judge is not suitable. How is the relationship of gurouronic acid can be a food supplement for skin protection? Currently, author used C6 not gurouronic acid. Besides, there is nor any evident experiments to show human digest system of C6 uptake which can relate to skin.

Response 3: Thanks very much for your suggestion. “Therefore, gurouronic acid, such as G6, may be a desirable food supplement for skin protection or the preparation of skincare products.” was corrected as “In vitro experiment results showed G6 had activated of the antioxidant system, which indicated the high potential of G6 as a candidate food supplement for skin protection or the preparation of skincare products.”

Point 4: Too many grammatical errors. Please refer the article to a native speaker for proof reading. Especially between each specific number and punctuation unit, a space is needed in the all article as well as the figures.

Response 4 : It’s so kind of you to remind us the spelling mistakes. We appreciated your help very much. We have check the article regarding grammar and spelling with the help of LetPub company.

 Point 5: Figure 4 (a), the TLC analysis is not a suit way.

Response 5: we are greatly appreciated your professional comments. On the one hand, our results suggested that G6 were successfully labeled with FITC. Additionally, As shown in (Figure 4(a)), As shown in Figure 4 (a), only the fluorescence of the FITC group is in the upper part of the plate, while the FITC-G6 line is only in the bottom of the silica gel plate, indicating that the successfully labeled FITC-G6 does not contain extra FITC molecules and will not affect the subsequent Interference in cellular experiments. On the other hand, other studies have used the same method to detect FITC labeled products.

Related research literature is as follows:

  1. Wang, X.; Jiang, H.; Zhang, N.; Cai, C.; Li, G.; Hao, J.; Yu, G., Anti-diabetic activities of agaropectin-derived oligosaccharides from Gloiopeltis furcata via regulation of mitochondrial function. Carbohydrate Polymers 2020, 229, 115482.
  2. Ravi, A.; Vijayanand, S.; Ramya, G.; Shyamala, A.; Rajeshkannan, V.; Aisverya, S.; Sudha, P.; Hemapriya, J., Alginates: Current Uses and Future Perspective. Alginates: Applications in the Biomedical and Food Industries 2019, 281-312.

Point 6: The resolution of Figure 4 (b) is too poor, the images are not convinced.Especially, in all of the DIC images, the cell seems not healthy?

Response 6: Thanks a lot for your helpful suggestions. We will submit the original image.

Point 7: Figure 5 (c). Why in the case of UVA+G6, the NAD+/NADH ratio is much higher than control?

Response 7: we are greatly appreciated your professional comments. Mitochondrial retrograde signaling starts with several main signals (such as ROS, ADP/ATP, and NAD+/NADH ratios), and ROS can lead to aging[65]. As shown in Figure 5(b)-(c), exposure to UVA radiation can reduce ATP generation and NAD+/NADH ratio. The pretreatment with G6 significantly reversed the UVA-induced decrease of ATP content and NAD+/NADH ratio, to the extent that the levels of these molecules were close to those of non-irradiated cells. In addition, these retrograde messengers activate cytosolic transducers by an oxidative modification that interact with small molecules, including Ca2+, NAD+, and AMP, which modulate the activities of certain transcription factors. These include pGC-1a, SIRT1, mTOR and CREB, which affect the mitochondrial and cellular function, namely the energy supply, and then change the progress of cellular senescence and proliferation[66-68]. SIRT1 is a member of the family of sirtuins that catalyze the deacetylation of various substrates by utilizing nicotinamide (NAD+) as a substrate[69].Furthermore, the NAD+/NADH ratio is much higher than control in the case of UVA+G6.

References:

  1. Finley, L. W.; Haigis, M. C., The coordination of nuclear and mitochondrial communication during aging and calorie restriction. Ageing research reviews 2009, 8, (3), 173-188.
  2. Butow, R. A.; Avadhani, N. G., Mitochondrial signaling: the retrograde response. Molecular cell 2004, 14, (1), 1-15.
  3. Chae, S.; Ahn, B. Y.; Byun, K.; Cho, Y. M.; Yu, M.-H.; Lee, B.; Hwang, D.; Park, K. S., A systems approach for decoding mitochondrial retrograde signaling pathways. Sci. Signal. 2013, 6, (264), rs4-rs4.
  4. Jazwinski, S. M.; Kriete, A., The yeast retrograde response as a model of intracellular signaling of mitochondrial dysfunction. Frontiers in physiology 2012, 3, 139.
  5. Kumar, R.; Nigam, L.; Singh, A. P.; Singh, K.; Subbarao, N.; Dey, S., Design, synthesis of allosteric peptide activator for human SIRT1 and its biological evaluation in cellular model of Alzheimer's disease. European journal of medicinal chemistry 2017, 127, 909-916.

Point 8: Discussion part is also very confused. Some descriptions are not refered to correct figure.

Response 8: It’s so kind of you to remind us the References mistakes. We have corrected and marked them in red. The number " Figure 4-5" in line 259 of the ninth page of the manuscript and the word "Figure 5-7" in line 277 have also been corrected.

Thanks for your advice again, and It’s hoped that our answers could give a clear explanation to your question.

Reviewer 2 Report

Thank you for your interesting article “Protective effect of G6 on UVA-induced photo-aging in HaCaT cells.”

The authors deal with the subject of photo ageing and to what extent this process and its consequences can be counteracted.

In the described research project the focus is on the investigation of G6 (guluronic acid hexaose), and its protective effects on photoaging in skin cells (secondary keratinocytes).

Using various molecular biological methods, it has been shown that G6 is localized next to the mitochondria and can improve mitochondrial functions. Oxidative stress and its consequences can be counteracted; in addition signaling pathways which are involved in preventing premature photo-aging damage can be activated.

The article is well and compact written and understandable for experts and also non-experts, and encourage the reader to read the whole article.

I support the publication of this interesting topic after a major revision.

Please check the article regarding grammar and spelling, sometimes the numbering (a,b..) is missing on the figures. The introduction part about G6, the main molecule in this article, is not introduced enough. It is not clear how this molecule was discovered. Reference [31] is not convincing. Please provide more background information. The term “ROS gene” (page 2, line 64) is too general. What kind of genes are meant here? Please describe in more detail. The scaling in Figure 2 a and b,should be changed, so that 1.0 is shown on the y-axis; thus the deviation from 100% can be detected faster by the reader. Has a positive control been carried along for the examination of cell viability? Please show these results to get a better assessment of toxicity. It is unusual to present the significance to a treated sample (UVA). The significance should be presented for the untreated sample (control), then the effect of irradiation and the positive effect of G6 is more impressive and traceable. The use of standard deviations must be chosen uniformly. Sometimes S.D. sometimes S.E.M was chosen; please uniformly! S.E.M. should not be selected to present minor errors -this is a falsification of the reader. In Figure 5a please change the scaling or show the y-axis in interruption so that the bars (next to UVA) can be distinguished better in their intensity. The reference in part 2.5 is wrong: refer to figure 6 a and b instead of figure 4-5. What chemical and physical properties does G6 have? In the material and methods section it is described that G6 is > 1000 Da. How was the uptake into the HaCaT cells accomplished? Are specific receptors for G6 given in the cell membrane? It is suggested by the scientists that G6 can be incorporated into skin care products or could be supplemented. It is well known that molecules > 500 Da hardly overcome the skin barrier..nothing is written about this. In general, the barrier of the stratum corneum (SC) represents the greatest challenge for the transport of active ingredients into the viable skin. Please add possibilities. Please describe in the material and method part the statistical analyses.

Author Response

Point 1: Please check the article regarding grammar and spelling, sometimes the numbering (a,b..) is missing on the figures.

Response 1: Thanks very much for your helpful comments. We have check the article regarding grammar and spelling with the help of LetPub company. We have made additions marked in red in the revised manuscript according to your suggestion. Added the numbering (a,b..) on figure 6 and 7.

Point 2: The introduction part about G6, the main molecule in this article, is not introduced enough. It is not clear how this molecule was discovered. Reference [31] is not convincing.

Response 2: Thanks very much for your suggestion. We have made additions marked in red in the revised manuscript according to your suggestion.

Other changes: 

(1). Line 2, the statement of “G6” was corrected as “L-Hexaguluroic Acid Hexasodium Salt”.

(2). Line 28, “Guluronic acid” was corrected as “L-Hexaguluroic Acid Hexasodium Salt”
(3). Line 406, “Guluronic acid hexaose” was corrected as “L-Hexaguluroic Acid Hexasodium Salt”.

(4). Line 76, “Marine alginate, a kind of acidic linear polysaccharide that consists of b-D-mannuronic acid and a-L-guluronic acid, has been reported widely due to its excellent biomedical activities[31].” was corrected as “Marine alginate is a natural anionic polysaccharide extracted from brown algae such as Laminaria japonica. It consists of  L-guluronic acid (G) and D-mannuronic acid (M) that occur in homopolymeric M blocks(M-blocks), heteropolymeric random MG blocks(MG-blocks),and homopolymeric G blocks(M-blocks). It has been widely  studied and applied in  chemical industry, medicinal technology and food engineering [31]. Moreover, alginate oligosaccharides produced by a variety of degradation methods exhibit excellent biomedical activities including anti-inflammatory activity, anti-oxidant activity and neuron protection effects[32-34].”.

Point 3: Please provide more background information. The term “ROS gene” (page 2, line 64) is too general. What kind of genes are meant here?

Response 3: It’s so kind of you to remind us the spelling mistakes. We appreciated your help very much. We have modified this section to modify ROS gene to ROS production according to Reference [21]. We have made additions marked in red in the revised manuscript according to your suggestion.

Point 4: The scaling in Figure 2 a and b, should be changed, so that 1.0 is shown on the y-axis; thus the deviation from 100% can be detected faster by the reader. Has a positive control been carried along for the examination of cell viability? Please show these results to get a better assessment of toxicity.

Response 4: Special thanks for your comments. Your suggestions are very valuable and helpful to our further study. We have changed the Figure 2 according to Reviewer's suggestion. The common positive control is vitamin C. Based on the results of our previous experiments, we selected G6 from several compounds with good activity.

Effects of G6  on cytotoxicity of HaCaT cells determined using MTT assays. The cells were pretreated with 10μM G1-G8 for 48h, and then irradiated with UVA(30J/cm2).Values are the mean of nine replicates ± S.E.M(n = 9). Significant difference compared to the UVA group, *P <0.05, **P <0.01, ***p < 0.001 and #P <0.0001.

The effects of G1-G8, Astragalus membranaceus  and Tremella on HaCaT cells viability were detected by MTT assays. And, the control group were normalized to be considered as 100%. The cells viability after Astragalus membranaceus and Tremella treatment was not significantly different from those in the control group, indicating that Astragalus membranaceus  and Tremella up to 200μg/ml had no significant cytotoxicity.Moreover,10μM (G1-G8)-pretreated were significantly protected HaCaT cells from the irradiation of UVA.

 Point 5:  (1) It is unusual to present the significance to a treated sample (UVA). The significance should be presented for the untreated sample (control), then the effect of irradiation and the positive effect of G6 is more impressive and traceable.(2) In Figure 5a please change the scaling or show the y-axis in interruption so that the bars (next to UVA) can be distinguished better in their intensity.

Response 5: It’s highly appreciated that you give us the suggestion to present the significance to the untreated sample (control). We have changed Figure2-8 according to Reviewer's suggestion.

Point 6: The use of standard deviations must be chosen uniformly.

Response 6: It’s so kind of you to remind us the spelling mistakes. We appreciated your help very much. We have modified this section to modify “mean ± SD (n = 9)” to “mean ±S.E.M (n = 9)” according to Reviewer's recommendations. We have made additions marked in red in the revised manuscript according to your suggestion.

Point 7: (1) The reference in part 2.5 is wrong: refer to figure 6 a and b instead of figure 4-5.(2) Please describe in the material and method part the statistical analyses.

Response 7: Thanks a lot for your helpful suggestions. We have made additions marked in red in the revised manuscript according to your suggestion in line164 and line 410-413.

Point 8: What chemical and physical properties does G6 have?

Response 8: Special thanks for your comments. We found the chemical and physical properties of G6 according to the instructions of Lantai Pharmaceutical Company (Qingdao, China)

The chemical and physical properties of G6 are as follows:

Formula:C36H44O37Na6                        Source: Marine brown algae

 FW:1206.65Da                     Storage:0~-20℃

 Purity:≥96%

Properties: Off-white to light yellow powdery or flocculent lyophilisate; odorless, non-irritating odor; hygroscopic

Point 9: In the material and methods section it is described that G6 is > 1000 Da. How was the uptake into the HaCaT cells accomplished? Are specific receptors for G6 given in the cell membrane?

Response 9: Thanks very much for your suggestion. Your suggestions are very valuable and helpful to our further study, More and more evidences showed that carbohydrates play essential and pivotal roles in a wide range of physiological and pathological process by the interaction between carbohydrate and protein (Chaudhary, Gade, Yellin, Sangabathuni, & Kikkeri, 2016; del Carmen Fernández-Alonso, Díaz, Alvaro Berbis, Marcelo, & Jimenez-Barbero, 2012; Fang et al., 2012; Qiu, Yang, Pei, Zhang, & Ding, 2010; S. Wang et al., 2009; Xiao et al., 2013). Generally speaking, carbohydrate protein interaction happens at specific site(Merritt et al., 1994) . For example, the aβ- 1,3/1,6-glucan (BG136) from Durvillaea Antarctica  can activate RAW264.7 cells by binding to TLR4 and then triggering TLR4-mediated signaling pathways to promote cytokine secretion(Yang et al., 2018); And, oral administration of GV-971(a novel marine-derived oligosaccharide) reconditions the gut microbiota, normalizes disordered metabolites, reduces the peripheral immune cell infiltration to the brain, resolves neuroinflammation, and reduces Aβ deposition and tau phosphorylation, leading to ultimate improvement of cognitive functions(X. Wang et al., 2019). Moreover, our present study (Figure 4) showed that G6 and mitochondria colocalize. As shown in Figure 7, residues of  Asp-204, Glu-416/208, Lys-203 and Arg-446 formed hydrogen bonds with one molecule of G6. Furthermore, these residues were also observed in the crystal structure of SIRT1/G6 complex with the 1:1 ratio, which interacted with G6 by hydrogen bonds. The observed results showed that it is likely that G6 bind within the SIRT1 pocket to produce a stable complex. Those previous works support our findings(Figure 7S).

We will continue to investigate the presence of G6 specific receptors in cell membranes based on current data.

REFERENCES

Chaudhary, P. M., Gade, M., Yellin, R. A., Sangabathuni, S., & Kikkeri, R. (2016). Targeting label free carbohydrate–protein interactions for biosensor design. Analytical Methods, 8(17), 3410-3418.

Damasceno, G. A., Barreto, S. M., Reginaldo, F. P., Souto, A. L., Negreiros, M. M., Viana, R. L., . . . Moura, R. A. (2020). Prosopis juliflora as a new cosmetic ingredient: development and clinical evaluation of a bioactive moisturizing and anti-aging innovative solid core. Carbohydrate Polymers, 115854.

del Carmen Fernández-Alonso, M., Díaz, D., Alvaro Berbis, M., Marcelo, F., & Jimenez-Barbero, J. (2012). Protein-carbohydrate interactions studied by NMR: from molecular recognition to drug design. Current Protein and Peptide Science, 13(8), 816-830.

Fang, J., Wang, Y., Lv, X., Shen, X., Ni, X., & Ding, K. (2012). Structure of a β-glucan from Grifola frondosa and its antitumor effect by activating Dectin-1/Syk/NF-κB signaling. Glycoconjugate journal, 29(5-6), 365-377.

Merritt, E. A., Sarfaty, S., Akker, F. V. D., L'Hoir, C., Martial, J. A., & Hol, W. G. (1994). Crystal structure of cholera toxin B‐pentamer bound to receptor GM1 pentasaccharide. Protein Science, 3(2), 166-175.

Qiu, H., Yang, B., Pei, Z.-C., Zhang, Z., & Ding, K. (2010). WSS25 inhibits growth of xenografted hepatocellular cancer cells in nude mice by disrupting angiogenesis via blocking bone morphogenetic protein (BMP)/Smad/Id1 signaling. Journal of Biological Chemistry, 285(42), 32638-32646.

Wang, S., Zhang, Z., Lin, X., Xu, D.-S., Feng, Y., & Ding, K. (2009). A polysaccharide, MDG-1, induces S1P1 and bFGF expression and augments survival and angiogenesis in the ischemic heart. Glycobiology, 20(4), 473-484.

Wang, X., Sun, G., Feng, T., Zhang, J., Huang, X., Wang, T., . . . Wang, H. (2019). Sodium oligomannate therapeutically remodels gut microbiota and suppresses gut bacterial amino acids-shaped neuroinflammation to inhibit Alzheimer’s disease progression. Cell research, 29(10), 787-803.

Xiao, F., Qiu, H., Zhou, L., Shen, X., Yang, L., & Ding, K. (2013). WSS25 inhibits Dicer, downregulating microRNA-210, which targets Ephrin-A3, to suppress human microvascular endothelial cell (HMEC-1) tube formation. Glycobiology, 23(5), 524-535.

Yang, Y., Zhao, X., Li, J., Jiang, H., Shan, X., Wang, Y., . . . Yu, G. (2018). A β-glucan from Durvillaea Antarctica has immunomodulatory effects on RAW264. 7 macrophages via toll-like receptor 4. Carbohydrate polymers, 191, 255-265.

Point 10: It is suggested by the scientists that G6 can be incorporated into skin care products or could be supplemented. It is well known that molecules > 500 Da hardly overcome the skin barrier..nothing is written about this. In general, the barrier of the stratum corneum (SC) represents the greatest challenge for the transport of active ingredients into the viable skin. Please add possibilities.

Response 10: we appreciate the reviewer's serious consideration of this issue. Previous studies showed the occurrence of polysaccharides and other natural products of interest for potential use as active and excipients ingredients in cosmetics, these include: saponins and phenolic compounds like flavonoids, tannins, and coumarins among others (Ibrahim, Nadir, Ali, Ahmad, & Rasheed, 2013; Rincón, Muñoz,Ramírez, Galán, & Alfaro, 2014). The use of polysaccharides similar to those described for P. juliflora with moisturizing and antioxidant activities has been reported and motivated the development of this work (Barreto et al., 2017). There is also the consumer perception that the use of plant-based raw materials is associated with safer and more environmentally friendly products (Antignac, Nohynek, Re, Clouzeau, & Toutain, 2011).It has been reported that the extract from Prosopis juliflora was fractioned and named F3M (molecular weight > 3KDa) and F3m (molecular weight <3KDa) and examine its application in a solid core formulation, which upon contact with water instantly forms a gel for immediate topical application as a moisturizing and anti-aging(Damasceno et al., 2020). Furthermore, we will focus on exploring the application of G6 in solid core formulations, increasing its potential as a skin care product.

REFERENCES

Antignac, E., Nohynek, G. J., Re, T., Clouzeau, J., & Toutain, H. (2011). Safety of botanical ingredients in personal care products/cosmetics. Food and Chemical Toxicology, 49(2),324-341.

Barreto, S. M. A. G., Maia, M. S., Benicá, A. M., de Assis, H. R. B. S., Leite-Silva, V. R., da RochaFilho, P. A., . . . Ferrari, M. (2017). Evaluation of in vitro and in vivo safety of the byproduct of Agave sisalana as a new cosmetic raw material: Development and clinical evaluation of a nanoemulsion to improve skin moisturizing. Industrial Crops and Products, 108(Supplement C), 470-479.

Ibrahim, M., Nadir, M., Ali, A., Ahmad, V. U., & Rasheed, M. (2013). Phytochemical analyses of Prosopis juliflora Swartz DC. Pakistan Journal of Botany, 45(6), 2101-2104.

Damasceno, G. A., Barreto, S. M., Reginaldo, F. P., Souto, A. L., Negreiros, M. M., Viana, R. L., . . . Moura, R. A. (2020). Prosopis juliflora as a new cosmetic ingredient: development and clinical evaluation of a bioactive moisturizing and anti-aging innovative solid core. Carbohydrate Polymers, 115854.

Thanks for your advice again, and It’s hoped that our answers could give a clear explanation to your question.

Round 2

Reviewer 1 Report

1) From author’s responds Based on our past experience in the laboratory experiments and platform regarding the research of drugs pharmacology, we screened several L-guluronic acid (G) activities, and then selected L-Hexaguluroic acid hexasodium salt with better activity for further mechanism research”, this part of discussion is highly recommended to be described in the article. Bur please confirm the correct grammatical writing. 

2) In the summary part, it is still hard to be convinced that “G6 can be a candidate food supplement for skin protection or the preparation of skincare products.” There is no any evidence of food supply having any related to skin. No evidence shows human digest system of C6 uptake which can relate to skin in current study.

3) The resolution of Figure 4’s images are still too low, although author provided the original images in a separated file. Please give a higher pixel image in the article.

4) Please add point 7’s reply in to the discussion part in the article. “ Mitochondrial retrograde signaling starts with several main signals (such as ROS, ADP/ATP, and NAD+/NADH ratios), a……Furthermore, the NAD+/NADH ratio is much higher than control in the case of UVA+G6.”

5) Some refs of protecting HaCaT Cell may need such as Int. J. Mol. Sci. 2019, 20(17), 4259.

Minor error:

1) In all of L-guluronic acid (G) and D-mannuronic acid (M) part, the symbol “L” and “D” are needed to have a smaller size typing.

2) Page 9, line 300. “G6 (Figure 1), with chemical and physical properties in Table1, was purchased …” There is space missing between Table 1.

3) Page 9, line 294. “ In vitro experiment results showed G6 had ….”

“In vitro” is needed to type with Italic style.

Author Response

Point 1: 1) From author’s responds “Based on our past experience in the laboratory experiments and platform regarding the research of drugs pharmacology, we screened several L-guluronic acid (G) activities, and then selected L-Hexaguluroic acid hexasodium salt with better activity for further mechanism research”, this part of discussion is highly recommended to be described in the article. Bur please confirm the correct grammatical writing. 

Response 1: We appreciate the reviewer’s serious consideration of this issue. We have made additions marked in red in the revised manuscript according to your suggestion.

It’s hoped that our answers could give a clear explanation to your question.

Point 2: 2) In the summary part, it is still hard to be convinced that “G6 can be a candidate food supplement for skin protection or the preparation of skincare products.” There is no any evidence of food supply having any related to skin. No evidence shows human digest system of C6 uptake which can relate to skin in current study.

Response 2: Thanks very much for your suggestion. “In vitro experiment results showed G6 had activated of the antioxidant system, which indicated the high potential of G6 as a candidate food supplement for skin protection or the preparation of skincare products” was corrected as “In vitro experiment results showed G6 had activated the antioxidant system, which indicated the high potential that G6 as a candidate supplement for skin protection or the preparation of skincare products.”

Point 3: 3) The resolution of Figure 4’s images are still too low, although author provided the original images in a separated file. Please give a higher pixel image in the article.

Response 3: Thanks a lot for your helpful suggestions. We have submit a higher pixel image in the article.

Point 4: 4) Please add point 7’s reply in to the discussion part in the article. “ Mitochondrial retrograde signaling starts with several main signals (such as ROS, ADP/ATP, and NAD/NADH ratios), a……Furthermore, the NAD+/NADH ratio is much higher than control in the case of UVA+G6.”

Response 4 : It’s so kind of you to remind us the spelling mistakes. We appreciated your help very much. We have made additions marked in red in the revised manuscript according to your suggestion.

 Point 5:5) Some refs of protecting HaCaT Cell may need such as Int. J. Mol. Sci. 2019, 20(17), 4259.

Response 5: we are greatly appreciated your professional comments. We have made additions marked in red in the revised manuscript according to your suggestion.

Point 6: Minor error:

1) In all of L-guluronic acid (G) and D-mannuronic acid (M) part, the symbol “L” and “D” are needed to have a smaller size typing.

2) Page 9, line 300. “G6 (Figure 1), with chemical and physical properties in Table1, was purchased …” There is space missing between Table 1.

3) Page 9, line 294. “ In vitro experiment results showed G6 had ….”

“In vitro” is needed to type with Italic style.

Response 6: It’s so kind of you to remind us the Minor error. We have corrected and marked them in red. (1-2)The symbol “L” and “D” in line77-78 of the manuscript and the space in line 305-306 have also been corrected.(3) The “In vitro” was corrected as

In vitro”.

Thanks for your advice again, and It’s hoped that our answers could give a clear explanation to your question.

Reviewer 2 Report

I am pleased that the proposals have been accepted and implemented and support now the publication of the article.

Author Response

On behalf of my co-authors, we thank you very much for giving us an opportunity to revise our manuscript. And the positive and constructive comments from editor and reviewers on our manuscript entitled “Protective effect of L-Hexaguluroic acid hexasodium salt on UVA-induced photo-aging in HaCaT cells”(ID: ijms-702159) were highly appreciated. These comments are all valuable and very helpful for revising and improving our paper, as well as the important guiding significance to our researches. We have studied comments carefully and have made correction which we hope to meet with approval. Revised portion are marked in red with underline in the paper. We would like to express our great appreciation to you and reviewers for comments on our paper again